# A Flexible Piezoelectric Device for Frequency Sensing from PVDF/SWCNT Composite Fibers

**DOI:** 10.3390/polym14214773

**Published:** 2022-11-07

**Authors:** Sejin Choi, Jihwan Lim, Hansol Park, Han Seong Kim

**Affiliations:** 1School of Chemical Engineering, Pusan National University, 2, Busandaehak-ro 63boen-gil, Geumjeong-gu, Busan 46241, Korea; 2Institute of Advanced Organic Materials, Pusan National University, 2, Busandaehak-ro 63boen-gil, Geumjeong-gu, Busan 46241, Korea

**Keywords:** piezoelectric, electrospinning, frequency sensing, flexible electronics

## Abstract

Polymer piezoelectric devices have been widely studied as sensors, energy harvesters, and generators with flexible and simple processes. Flexible piezoelectric devices are sensitive to external stimuli and are attracting attention because of their potential and usefulness as acoustic sensors. In this regard, the frequency sensing of sound must be studied to use flexible piezoelectric devices as sensors. In this study, a flexible piezoelectric device composed of a polymer and an electrode was successfully fabricated. Polyvinylidene fluoride, the active layer of the piezoelectric device, was prepared by electrospinning, and electrodes were formed by dip−coating in a prepared single−walled carbon nanotube dispersion. The output voltage of the external sound was matched with the input frequency through a fast Fourier transform, and frequency matching was successfully performed, even with mechanical stimulation. In a high−frequency test, the piezoelectric effect and frequency domain peak started to decrease sharply at 300 Hz, and the limit of the piezoelectric effect and sensing was observed from 800 Hz. The results of this study suggest a method for developing flexible piezoelectric-fiber frequency sensors based on piezoelectric devices for acoustic sensor systems.

## 1. Introduction

Acoustic detectors are used in a wide range of fields, such as voice communication, environmental noise monitoring, flight tracking and detection, and photoacoustic spectroscopy [1,2,3]. In addition, many frequency-vibration sensors have been used over the years in various ways as accelerometers. For example, they have been used to supervise conditions in heavy industries and in oil-pipeline vibration monitoring and structural condition monitoring. They are also used in the detection of tectonic motions, even motions due to earthquakes, volcanoes, landslides, etc. [4,5,6,7]. With the harvesting of mechanical vibration energy using the triboelectric effect, research on sensing vibrations generated in various areas, such as general vibrations, rotational motion, and wind and water flow, has been actively conducted [8,9,10,11]. In particular, various studies are being conducted on fiber-based sensors, such as Bragg gratings and tapers for the detection of mechanical and acoustic vibrations. With the expectation of their application in various fields due to the flexible characteristics of the fibers, interest in next-generation frequency sensors based on various sensing mechanisms using functional materials is increasing [12,13,14]. Recently, piezoelectric devices that convert mechanical energy into electrical energy or signals have gained much attention [15,16,17]. In particular, with the expansion of wearable electronic devices, flexible piezoelectric devices are attracting attention as flexible energy harvesters, sensors, and generators, amongst other applications [18,19,20]. In this regard, many studies have focused on piezoelectric performance, sensor utilization, and generation efficiency [21,22,23]. Polymeric piezoelectric materials must be used to develop flexible piezoelectric devices. Ceramic piezoelectric materials have been reported to exhibit stronger piezoelectric effects than polymer materials [24]. However, owing to the brute properties of ceramics, their applications are limited, and acoustic piezoelectric devices based on vibration energy have been reported to have close relationships with device performance [25,26]. In this context, flexible piezoelectric acoustic sensors have been highlighted for potential high-value-added applications, such as cochlear devices [27,28]. Accurate sensing of sound frequency is important for piezoelectric acoustic sensors, and more flexibility is required for accurate responses [27,29]. Poly(vinylidene fluoride) (PVDF) is a piezoelectric polymer material that is easily accessible, inexpensive, and easy to process. Although the material has a lower piezoelectric constant than ceramic piezoelectric materials, its inherent flexibility and softness make it highly suitable for acoustic sources [30,31]. The piezoelectric effects of PVDF are influenced by the β-phase crystal structure of the polymer [32,33,34]. There are various techniques for inducing β-phase crystals in PVDF, including casting, spin coating, mechanical stretching, poling, and electrospinning. In particular, electrospinning can be applied for mechanical stretching by fiber elongation and high-voltage poling [35,36,37,38]. In this paper, we present a flexible acoustic frequency-sensing device. Fiber-type PVDF was prepared by electrospinning, and the electrodes were prepared by dip-coating in an SWCNT dispersion. The matching according to various acoustic frequencies was evaluated, and the piezoelectric and sensing limits due to mechanical forces were investigated. Although the fiber-type PVDF showed a low output voltage, it achieved frequency matching through a fast Fourier transform (FFT). These results suggest that it could be used as a flexible part of piezoelectric acoustic sensors.

## 2. Materials and Methods

### 2.1. Materials

PVDF powder (Mw = 534,000; Sigma–Aldrich Inc., Saint Louis, MO, USA) was added to a solvent mixture of N,N-dimethylformamide (DMF; Junsei Chemical Co., Ltd., Tokyo, Japan) and acetone (Duksan Pure Chemicals Co., Ltd., Ansan, Korea). The DMF/acetone ratio was 5:5, and the PVDF concentration was 14 wt%. After preparing a 14 wt% PVDF solution and a dispersion of 0.04 wt% single-walled carbon nanotubes (SWCNTs) with the same weight, the prepared solution and dispersion were mixed with additional PVDF powder to obtain a 15 wt% PVDF solution with 0.02 wt% SWCNTs. To produce flexible electrodes, SWCNTs (Tuball Single Wall Carbon Nanotubes, Ocsial, Luxembourg) were dispersed in water at 0.1% with 0.2 wt% sodium dodecylbenzene sulfonate (SDBS; Sigma–Aldrich Inc., USA) using a tip sonicator.

### 2.2. Fabrication

We fabricated PVDF/SWCNT webs, which are flexible piezoelectric devices, via electrospinning. The PVDF/SWCNT solution was fed through a 21 G metal nozzle (inner diameter = 0.495 mm) at a pressure of 20 kPa using compressed air to maintain a constant feeding rate. The nozzle and solution were applied at a high voltage of 9.5 kV. The tip-to-collector distance (TCD) was 15 cm. The resulting PVDF/SWCNT webs were collected for 6 min at room temperature (25 °C) and a relative humidity of 50% and subsequently dried for one day. Then, the fabricated PVDF/SWCNT webs were coated with the SWCNT dispersion by the dip-coating method to form flexible electrodes.

### 2.3. Measurement

The morphologies of the piezoelectric devices (the SWCNT-coated PVDF/SWCNT webs) were observed using field-emission scanning electron microscopy (FESEM; SUPRA 25, Carl Zeiss Co., Ltd., Oberkochen, Germany) to confirm the conditions of the spun and SWCNT-coated PVDF/SWCNT webs. Using a commercial speaker (XS-XB 160p, Sony, Tokyo, Japan), sound waves with fixed pressures and different frequencies were generated with a PVDF/SWCNT device fixed on a supporting frame. The sound pressure and frequency were controlled by a voltage and various input frequencies with a function generator (AFG2225, GWInstek, New Taipei, Taiwan) connected to the speaker. The frequency range of the function generator was 10–100 Hz, and the input voltage was 1 V. The output voltages from the piezoelectric device were measured using a data-acquisition system (USB-6210, National Instruments, Austin, TX, USA) with an external load resistance of 200 MΩ. To evaluate the frequency-sensing limits of the piezoelectric element with respect to the input frequency, a smart shaker (K2004E01, MODAL SHOP, Cincinnati, OH, USA) was used to generate a periodic load source (Figure 1). In order to apply a 200 gf constant load, a load cell was installed under the smart shaker to adjust the distance from the shaker to apply a 200 gf load (Figure 1). A function generator (AFG-2225, GWInstek, Xinbei, Taiwan) was used to control the frequency of the shaker, which was increased from 100 to 1200 Hz at an input voltage of 1 V at 50 Hz. The same output-signal acquisition method was used in this study. The matching between the frequency domain and the input frequency was analyzed through FFT analysis of piezoelectric data using external sound.

## 3. Results and Discussion

The bead-in-fiber formation of PVDF webs fabricated by electrospinning has been reported to be an important factor that interferes with the piezoelectric effect [13]. As shown in Figure 2, the PVDF webs formed in random orientations without beads and had an average diameter of 1.83 ± 1.22 μm. As shown in Figure 2b, the morphological analysis of the coated PVDF webs showed that the SWCNTs formed between the fibers in thin-film-like shapes and that the porous structure was maintained. It is considered that the partial rather than a complete coating was due to the large difference in the surface energies of PVDF and water as a solvent in the SWCNT dispersion. Moreover, the SWCNT films were connected as a network; thus, the SWCNT films can serve as electrodes.

During electrospinning, a β-phase crystal of the PVDF fiber webs formed through fiber poling and mechanical elongation. Figure 3a shows the FTIR spectrum of the PVDF fiber webs. The characteristic absorption peaks of the β-phase were observed at 840 and 1274 cm^−1^, while the α-phase crystal peaks appeared at 763 and 975 cm^−1^. The β-phase crystal peaks were relatively more intense than the α-phase crystal peak, and the relative crystal fraction of the β-phase crystal could be calculated using the following equation [39,40]:(1)$$F\left(\beta \right)=\frac{A_{\beta }}{\left(K_{\beta }/K_{\alpha }\right)A_{\alpha }+A_{\beta }}$$
where *A*_α_ and *A*_β_ represent the absorbencies of each phase and *K*_α_ and *K*_β_ represent the absorption coefficients at each wavenumber. The relative β-phase fraction of the PVDF webs in this study was calculated to be 0.73. The characteristic XRD pattern peaks appeared at 18.4°, 20.1°, and 26.7° for the α-phase crystal diffraction and at 20.5° for the β-phase crystal diffraction [40,41]. As shown in Figure 3b, the β-phase crystal peak was observed without the α-phase crystal peaks of the electrospun PVDF webs, suggesting that the fabricated PVDF webs had sufficient β-phase crystals for use in piezoelectric devices.

The characteristics of the piezoelectric outputs with respect to the generation times for the sound waves were investigated. Sound waves of various frequencies were applied to the PVDF webs, and the frequencies of the sound waves were controlled by a function generator. As shown in Figure 4a, the time−voltage graph shapes changed in various forms according to the input frequency (sound frequency) and were similar to the sound waveforms. The non−woven piezoelectric device is considered to vibrate in the same way as the input sound wave owing to its flexible structure and thus exhibited output voltages similar to the sound−wave shapes. Although the differences in the maximum output voltages did not appear according to the input frequencies, the waveforms of the time−voltage graphs changed as the input frequencies changed. In particular, the graphs for 10, 50, and 70 Hz and for 20, 40, and 80 Hz respectively showed similar waveforms. Figure 4b shows the results of processing the time−voltage data. With the FFT results for 10, 50, and 70 Hz and for 20, 40, and 80 Hz, maximum peaks were observed in the same frequency domains as the frequencies corresponding to each input frequency. At 10, 50, and 70 Hz and at 20, 40, and 80 Hz, corresponding low peaks appeared as similar graph waveforms, and a peak was observed at 60 Hz at all frequencies, which was thought to be the result of noise caused by 220 VAC and 60 Hz−electrical standards used in experimental equipment and computers. Although the time−voltage graphs did not show intuitive dependences on input frequencies, they appeared as composite results of other frequencies, such as 60 Hz noise. Accordingly, it can be seen that there was no change in output voltage according to input frequency, while the graph waveforms changed with respect to time−voltage changes and the peaks according to the FFT processing for these changes.

Figure 5 shows the results of the FFT processing of the time−voltage data in Figure 4 to investigate the sound frequency sensing performance of the flexible piezoelectric device. In Figure 5a, fixed peaks caused by the constant noise from external devices were observed in the 60 Hz region. The maximum peaks in the frequency domain matched the input frequencies. In Figure 5b, the frequency domain and input frequency matched for sounds with input frequencies of 76, 77, 78, and 79 Hz, with a difference of 1 Hz. Hence, the flexible piezoelectric device could be fully utilized as an acoustic frequency sensor with a resolution of 1 Hz.

Figure 6a shows the output voltages and applied loads at 10 Hz and 100 Hz, and it can be seen that the output voltage appeared according to the load that matched the frequency. Figure 6b shows the piezoelectric output voltages for various frequencies with a load of 200 gf. These voltages allowed us to evaluate the performance for direct mechanical force and sound−frequency sensing. Relatively high voltages were outputted due to large deformations caused by direct stronger forces that were stronger than those caused by sound, and the trend of the waveform’s period changed at 30 Hz, like that of sound. However, above 90 Hz, it was difficult to identify the form characteristics using the graphs.

To investigate the sound−frequency sensing and piezoelectric limits of the flexible piezoelectric device, we measured piezoelectricity using a high−frequency load of 200 gf. Figure 7a shows the output voltage and applied load at 200 Hz, and it can be seen that the output voltage appeared according to the load that matched the frequency. It can be seen that even at a very high frequency of 1200 Hz, a load of about 200 gf was applied, but the output voltage did not appear. At an input frequency of about 300 Hz, the non−woven piezoelectric element was re−loaded before it was completely deformed and recovered by the applied load, so it was judged to be significantly lower than the piezoelectric output voltage at a frequency under 300 Hz. At the input frequency of 600 Hz, the voltage output by the applied load showed a very weak output voltage from noise. From 800 Hz, only noise was measured, and no characteristic results were observed. In the limit analysis of acoustic−frequency sensing through FFT in Figure 7c, the peak of the frequency domain matching the input frequency rapidly reduced by 300 Hz, which was in line with the result shown in Figure 7b. With an input frequency of 800 Hz, the peaks were lower than the unmatched peaks or were not observed. Considering this, the input frequency of 300 Hz showed the limit of the available piezoelectric performance, and it is considered that the frequency sensing limit is 600 Hz. This limitation exists because the piezoelectric effect due to deformation and recovery cannot reverse the deformation owing to an external stimulus that is too fast, and the deformed state is maintained.

Table 1 compares the relative characteristics of the vibration frequency sensing device devised in this study and other vibration frequency sensors. Although there have been many reports of sensors with similar characteristics, we have focused on the mode and type of operation of the sensor. Most sensors have different frequency detection ranges. However, there are few sensors in the form of fabrics and fabrics that are easy to process and which can be applied in various fields. We investigated sensor devices ranging from low to relatively high frequencies as fiber−based, fabric−type sensors. The vibration−frequency−detection sensor element proposed in this study is in the form of a fiber−based fabric produced through electrospinning and has high flexibility, so it can show a relatively high-frequency response by responding immediately to vibration. It is also expected to operate in extreme environments that can induce high strain, unlike ceramic−based or packaged sensors.

## 4. Conclusions

The performance and performance limitations of acoustic−frequency sensing were investigated through frequency matching for a flexible piezoelectric device. For the flexible piezoelectric device, a PVDF fiber piezoelectric layer was manufactured through electrospinning, and SWCNTs were manufactured as electrodes by a dip−coating method. The waveforms of the time–voltage curves changed according to the frequencies of the external sound, and, through FFT analysis, the frequency domains and external sound frequencies were matched. The limitations of piezoelectric and acoustic−frequency sensing were evaluated by applying mechanical vibrations at high frequencies, and we confirmed that the acoustic−frequency sensor exhibited sufficiently high performance. These results suggest that it is simple to manufacture and use PVDF fibers for application in high−performance, polymer−based, flexible piezoelectric frequency sensors.

## Figures and Tables

**Figure 1 polymers-14-04773-f001:**
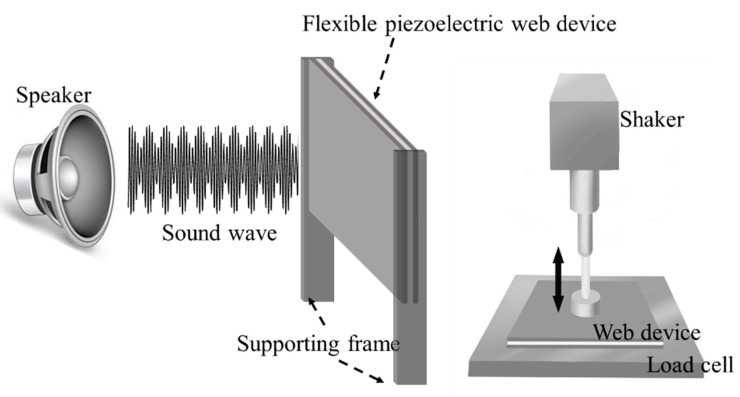
Schematic illustration of the piezoelectric setup for sound waves and mechanical forces.

**Figure 2 polymers-14-04773-f002:**
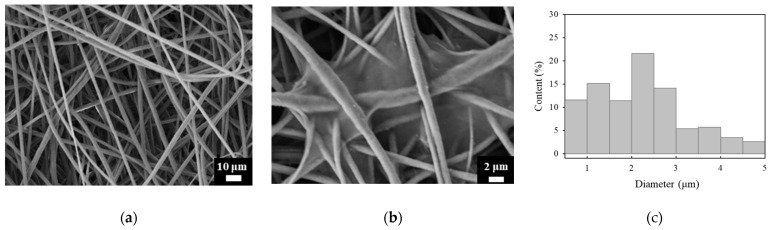
(**a**) FE-SEM image of the PVDF webs before and (**b**) after dip-coating. (**c**) Fiber diameter distribution.

**Figure 3 polymers-14-04773-f003:**
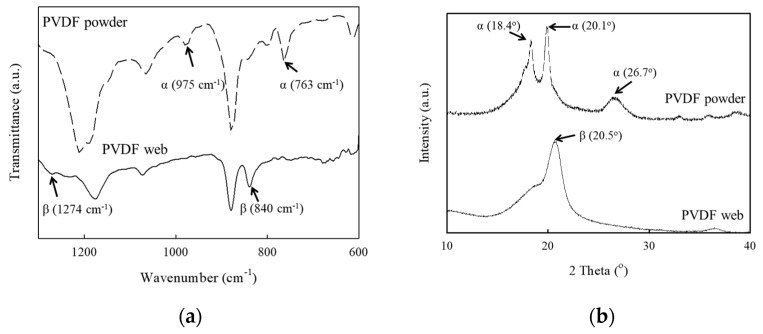
(**a**) FTIR spectra and (**b**) XRD patterns of the electrospun PVDF webs and PVDF powder.

**Figure 4 polymers-14-04773-f004:**
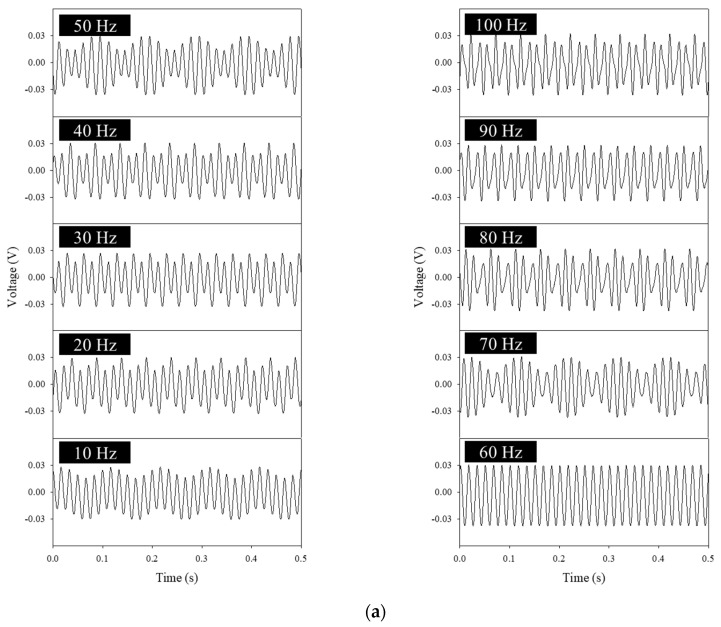
(**a**) Acoustic output voltages of the PVDF webs according to frequencies. (**b**) Fast Fourier transform (FFT) according to 10, 50, and 70 Hz and to 20, 40, and 80 Hz.

**Figure 5 polymers-14-04773-f005:**
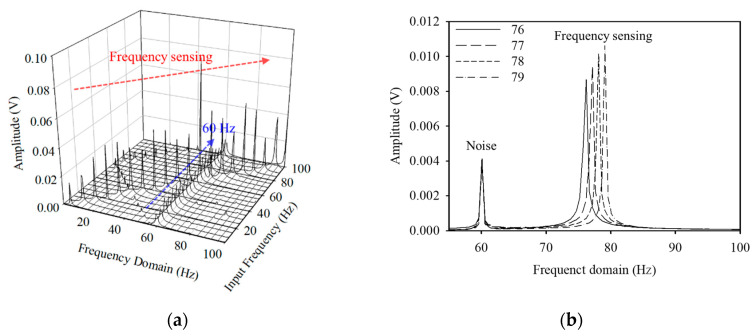
(**a**) Fast Fourier transform (FFT) matching according to the input frequencies and (**b**) narrow ranges of frequencies.

**Figure 6 polymers-14-04773-f006:**
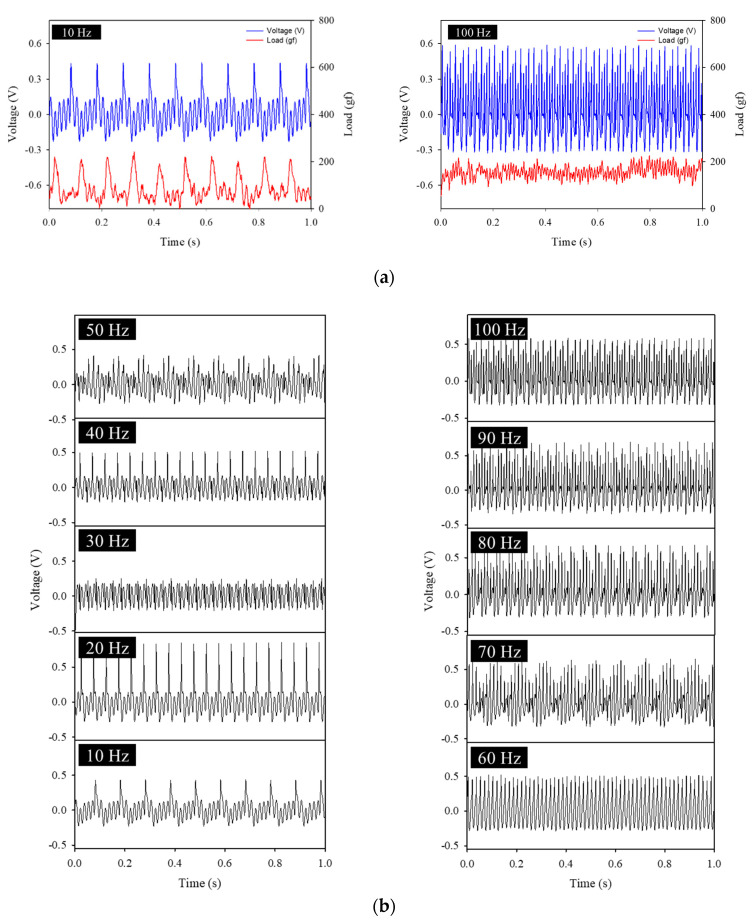
(**a**) Output voltages and applied loads for 10 and 100 Hz. (**b**) Output voltages of mechanical forces applied to the flexible piezoelectric device according to the frequency at 200 gf.

**Figure 7 polymers-14-04773-f007:**
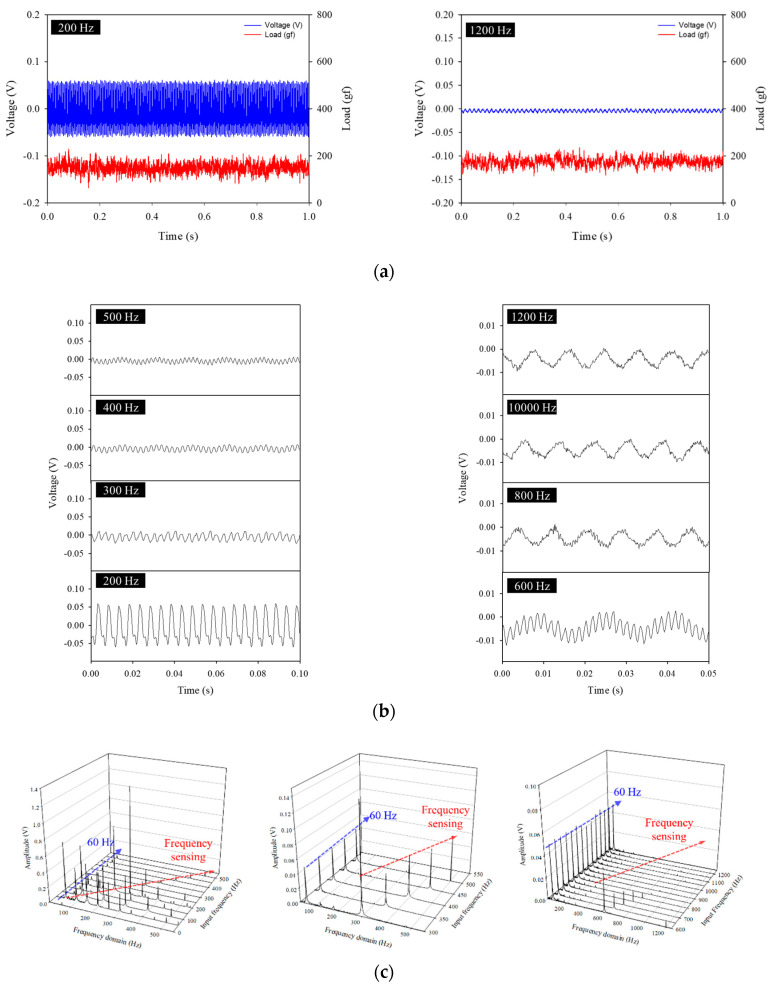
(**a**) Output voltages and loads for 200 and 1200 Hz. (**b**) Output voltages by mechanical forces applied to the flexible piezoelectric device according to high frequencies at a load of 200 gf. (**c**) FFT matching according to high−frequency inputs.

**Table 1 polymers-14-04773-t001:** Comparison of vibration−frequency sensors.

Sensor	Type	Processability	Frequency Range	Ref.
Cylinder	Triboelectric	Complex	10–50 Hz	[8]
			0–120 Hz	[41]
MEMS	Capacitive	Complex	100 Hz–10 kHz	[42]
Rod	Piezoelectric	Simple	80–130 Hz	[43]
Textile	Piezoelectric	Very simple	~600 Hz	This study

## Data Availability

The data presented in this study are available on request from the corresponding author.

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
