# Peer review of "A Flexible Piezoelectric Device for Frequency Sensing from PVDF/SWCNT Composite Fibers"

_polymers, 2022, doi:10.3390/polym14214773_

Round 1
Reviewer 1 Report
1. The experiment setup needs to be shown clearly.
2. The sound pressure sensed by the acoustic pvdf sensor is different due to the acoustic resistance of air to different acoustic signal. So extra lines need to be added to analyzed the local sound pressure near the pvdf cantilever beam.
3. In principle, the use of pvdf sensor to detect sound is nothing new. The authors need to show the advantages and novelties of the current contribution.
4. The vibration shaker is said to provide a signal with 20gf amplitude and frequency ranging from 100 to 1200 Hz. This is experimentally impossible because of the power limit of the vibration shaker itself. So it is doubted that especially in higer frequency range, the actual load amplitude might not be 20gf, and might be much smaller. Results and discussions should be given for this point.
Author Response
첨부파일을 참조하시기 바랍니다.

Reviewer 2 Report
In this work, the authors reported PVDF based flexible piezoelectric device investigated the performance of acoustic frequency sensing. After carefully reading the paper, it’s found that the paper is roughly prepared. There is no dependence of input sound frequency on output voltage frequency in Figure 4. Figure 4 is confusing and contrary with Figure 5. Figure 7a is also confusing, from which no signals can be obtained above 200 Hz and the signals in 200 Hz is not convincing because of the large time scale of 0.5 s. However, the authors concludes that the limitation of sensing frequency in their devices is 800 Hz. I suggest the paper should not be published at Polymers journal.
Round 2
Reviewer 1 Report
accept
Author Response
We appreciate the time and effort of the referee in reviewing our manuscript and their positive comments on the review report. Minor revisions of the Academic Editor are indicated in the revised manuscript.
Reviewer 2 Report
The author's response has satisfied me, I suggest it's publiction.
Author Response

(The authors gave the same response as above.)
